# China’s Vaccine Diplomacy and Its Implications for Global Health Governance

**DOI:** 10.3390/healthcare10071276

**Published:** 2022-07-10

**Authors:** Liangtao Liu, Yongli Huang, Jiyong Jin

**Affiliations:** 1Honors College, Shanghai International Studies University, Shanghai 200083, China; 0203100008@shisu.edu.cn; 2School of Journalism and Communication, Shanghai International Studies University, Shanghai 200083, China; yonglih@shisu.edu.cn; 3School of International Relations and Public Affairs, Shanghai International Studies University, Shanghai 200083, China

**Keywords:** China, COVID-19, vaccine diplomacy, health security

## Abstract

The COVID-19 pandemic has wreaked havoc on global economy and human communities. Promoting the accessibility and affordability of vaccine via diplomacy is the key to mitigating the pandemic crisis. China has been accused of seeking geopolitical objectives by launching vaccine diplomacy. The definition of vaccine diplomacy is neutral by nature. China’s vaccine diplomacy is based on its holistic approach to national security and the importance China attaches to the “Belt and Road” Initiative. With a whole-of-government approach on both the bilateral and multilateral levels and marketization of vaccines, China’s vaccine diplomacy has immense implications for global health governance, in that it helps to narrow the global immunization vaccination gap and to promote human-right-based approach to global health governance. However, the sustainability of China’s vaccine diplomacy is questionable because of the Sino-American geopolitical competition and doubts over the efficacy of China’s vaccines. The escalation of power rivalry between China and the U.S. and the concerns over the efficacy of China’s vaccines forebode the gloomy future of China’s vaccine diplomacy.

The COVID-19 pandemic has wreaked havoc on the world economy and global health security. To some degree, China has successfully developed vaccines to tackle the pandemic. However, many countries in the world, particularly the least-developed countries, lack the biotechnology to develop vaccines. No country will be safe until all countries are safe in terms of pandemic control. Therefore, it is both morally and realistically imperative for China to promote accessibility, affordability and availability of vaccines for the international community. China temporarily put the pandemic under control domestically and offered vaccines both within and beyond its borders. However, China has been accused of conducting vaccine diplomacy to expand its influence and achieve geopolitical objectives. This paper aims to explore the concept of China’s vaccine diplomacy in the context of COVID-19 and to identify China’s motivations to launch vaccine diplomacy and its implications for global health governance. The challenges of China’s vaccine diplomacy ahead are also examined in the paper. 

## 1. Vaccine Diplomacy and China

### 1.1. Vaccine Diplomacy

Vaccine diplomacy is not something new. It is as old as the vaccine itself [1]. Vaccine diplomacy has been launched to promote international health and international relations. Hotez defines vaccine diplomacy as “any aspect of global health diplomacy that relies on the use or delivery of vaccines” [2] (p. 43). Shakeel et al. describe vaccine diplomacy as the branch of global health diplomacy “that promotes the use and delivery of vaccines to achieve larger global health goals and shared foreign policy objectives” [3] (p. 82). Vaccine diplomacy contributes to global health security. The U.S.–Russia cooperation on vaccines to eradicate smallpox worldwide in the 1970s is a remarkable example of successful vaccine diplomacy [4] (p. 1301).

Vaccine diplomacy involves research, development, production and exchange of vaccine products in health diplomacy. It refers to both the means and ends in international cooperation on vaccines. It is neutral by nature, as it is regarded as an end to promote the availability, accessibility and affordability of vaccines via diplomacy. Meanwhile, it could be used as a means to achieve diplomatic and political objectives. For example, the Central Intelligence Agency (CIA) of the U.S. once used vaccination programs for intelligence purposes [5] (p. 413). As another example, the U.S. government used vaccines as a means to win goodwill from the Indians in the west region and other countries in the 19th century [6].

### 1.2. Doubts over China’s COVID-19 Vaccine Diplomacy

China has launched unprecedented vaccine diplomacy since vaccines against the COVID-19 have become available in April of 2020. However, China’s vaccine diplomacy has incurred widespread criticism. Eckart Woertz and Roie Yellinek claim that vaccine diplomacy has entered China’s political dictionary. China uses this means to coerce recipient countries into conducting business [7]. Some argue that China uses vaccines to increase its influence and presence in certain areas to confront and compete with the U.S. [8]. Additionally, China has been accused of manipulating vaccine narratives in its competition with the United States. [9]. *The Financial Times* warned the Western countries of the increasing influence of China with its vaccines [10]. China was also criticized on the ground that its vaccine diplomacy undermined regional prosperity [11]. The American Congress even released the Curbing China’s Vaccine Diplomacy Act to respond to China’s vaccine diplomacy [12]. The doubts cast over China’s vaccine diplomacy make it worthwhile to investigate China’s motivations for prioritizing vaccines in its diplomatic agenda.

## 2. China’s Motivations for Launching Vaccine Diplomacy

China has actively engaged itself in vaccine diplomacy since the outbreak of the COVID-19 pandemic. China’s vaccine diplomacy derives from its holistic approach to national security and commitment to building the Health Silk Road.

### 2.1. China’s Pursuit of a Holistic Approach to National Security

Vaccine diplomacy is a means for China to achieve its national security. China adopts a holistic approach to national security, as biosecurity has been integrated with its overall national security strategy [13]. In June 2020, Xi Jinping stressed the importance of building a strong public health system to provide adequate support for people’s health in such fields as reform of the disease control and prevention system, improvement of the legal system related to health and promotion of international cooperation [14]. The Biosecurity Law of the People’s Republic of China was released by China’s National People’s Congress in October 2020. This law “is formulated so as to maintain national security, prevent and respond to biosecurity risks, safeguards people’s lives and health, protect biological resources and the ecological environment, promote the healthy development of biotechnology, promote the construction of a community with a shared future for mankind” [15] (pp. 2–3). It specifies that biosecurity is an important part of national security, and China will promote international cooperation in biosecurity [15] (pp. 3–4). The law also states that China will mitigate biosecurity threats posed by the emerging infectious diseases [15] (p. 21). Vaccine diplomacy is an indispensable part of international cooperation in biosecurity. China’s unprecedented formulation of the Biosecurity Law explicitly speaks volumes of China’s efforts to address biosecurity threats for its overall national security. In addition, China’s speech acts on COVID-19 pandemic and the extraordinary measures China adopted to address the pandemic indicate that health security has been combined with China’s holistic national security framework. No country is safe from COVID-19 until every country is safe. The biosecurity interdependence between China and other countries motivated China to implement vaccine diplomacy for its national security.

### 2.2. China’s Commitment to Building a Health Silk Road (HSR) for Cooperation

The concept of the HSR was proposed by China’s National Health Commission in 2015. At the HSR Construction Symposium in 2016 in Uzbekistan, Xi Jinping stressed the importance to work jointly with other countries to build the HSR [16]. China committed itself to building the Health Silk Road by promoting cooperation in the public health system among countries along the Belt and Road routes [17] (p. 3). China even signed a Memorandum of Understanding on Health Cooperation within the framework of the Belt and Road Initiative with the World Health Organization (WHO) in 2017 [18].

The HSR is set to further advance China’s role in global health governance. The COVID-19 pandemic has presented China a wonderful opportunity to build the HSR by practicing vaccine diplomacy in its engagement in global health governance. Having cast vaccines as international public goods, China has taken steps to provide vaccines or localize the production of vaccines in the participant countries of the HSR initiative as a means of bridging the immunization gap. For example, China launched the Initiative for Belt and Road Partnership on COVID-19 Vaccines Cooperation with 28 countries, including Kazakhstan, Thailand and Colombia, and called for international cooperation on vaccines [19]. China has directly provided vaccines to four geographical regions. Out of these four regions, Asia Pacific, the preferred region for HSR construction, has received the most significant number of Chinese vaccines. Not surprisingly, 9 out of the top 10 biggest recipients of China’s donated doses of vaccine are participant countries of the HSR initiative.

## 3. China’s Approaches to Vaccine Diplomacy

### 3.1. China’s Whole-of-Government Approach to Vaccine Diplomacy

China adopts a whole-of-government approach to vaccine diplomacy. Effective vaccine diplomacy entails close coordination among various ministries of the Chinese government. Three ministries play a key role in China’s vaccine diplomacy: China International Development Cooperation Agency (CIDCA), Ministry of Commerce (MOC) and Ministry of Foreign Affairs (MFA). In general, CIDCA is responsible for formulating foreign aid guidelines and policies. MOC and MFA are in charge of implementing vaccine diplomacy. The former is also responsible for the specific implementation of foreign aid and negotiation with aid recipients and handling of specific affairs related to foreign aid projects, while MFA’s agencies abroad, such as embassies and counsels, are taking the responsibility to coordinate and manage the foreign aid in the host country [20].

China’s vaccine diplomacy also involves many other ministries or agencies due to the urgency and complexity of vaccine production and distribution, such as the Ministry of Industry and Information Technology (MIIT), National Health Commission (NHC), Ministry of Transport (MOT), Ministry of Finance (MOF), General Administration of Customs (GAC), National Medical Products Administration (NMPA), etc. [21]. The coordination among so many ministries exemplifies China’s whole-of-government approach to vaccine diplomacy. 

### 3.2. China’s Vaccine Diplomacy on Both Bilateral and Multilateral Levels

China has launched its vaccine diplomacy on bilateral and multilateral levels. China’s Sinopharm and Sinovac are the main providers of vaccines abroad. The immunization inequality is widening between the developed and developing countries. As of August 2021, around 60% of the population of higher-income countries had received at least one dose of the coronavirus vaccine. By contrast, only 1% of poorer populations in lower-income economies had received at least one dose of a vaccine at the same time point [22] (p. 1). China committed itself to narrowing the immunization gap. At the Global Health Summit, China’s President Xi Jinping reiterated the urgency to find solutions to issues concerning the production capacity and distribution of vaccines in order to make vaccines more accessible and affordable in developing countries. He called on the international community to uphold fairness and equity to close the immunization gap [23]. China pledged to offer 1 billion additional doses of vaccines to Africa, among which 600 million doses will be provided as donation, and 400 million doses will be provided through joint production by Chinese companies and relevant African countries [24]. With regard to ASEAN countries, China promised to donate 150 million doses and provide USD 5 million for the COVID-19 ASEAN Response Fund [25].

Apart from bilateral cooperation in donations and sales of vaccine, China’s vaccine diplomacy is extended to bilateral joint research and development of vaccines. For example, China and Cuba jointly developed a vaccine called Pan-Corona. The research was conducted in China and led by specialists from Cuba [26]. China has cooperated with Serbia to produce a China-developed COVID-19 vaccine [27]. At the beginning of 2022, China’s Sinovac and the Egyptian government agreed to accelerate the transfer of COVID-19 vaccine production technology and the building of a warehouse with a capacity for 150 million vaccine doses [28]. China also collaborated with Pakistan to produce the Chinese CanSinoBio COVID-19 vaccine [29]. China’s efforts on the bilateral level to produce vaccines contributed to the accessibility and affordability of vaccines in the developing countries. 

China adheres to multilateralism in its foreign policy. China’s vaccine diplomacy has also been implemented on the multilateral level. At the International Forum on COVID-19 Vaccine Cooperation hosted by China on 6 August 2021, Xi Jinping announced that “we (China) stand ready to work with international organizations to advance vaccine cooperation to protect the international community for a shared future” [30]. China’s multilateralism in vaccine diplomacy finds expression in its contribution to COVAX, a multilateral international COVID-19 vaccine initiative led by the World Health Organization. COVAX is the main mechanism for China’s vaccine diplomacy at the global level. China has provided over 180 million doses of vaccines to 49 countries through COVAX [31]. China’s multilateralism in vaccine diplomacy is also reflected in its collaboration with Gavi, a multilateral mechanism dedicated to improving vaccine accessibility in lower-income countries. China hosted the International Forum on COVID-19 Vaccine Cooperation on 6 August 2021. At the forum, China pledged USD 100 million to the Gavi COVAX Advance Market Commitment to finance equitable access in 92 lower-income countries [32]. The announcement is China’s largest voluntary pledge to an international organization to date. On 12 July 2021, China’s Sinopharm and Sinovac signed agreements with Gavi to provide 550 million doses of vaccines [33]. China’s contribution to the aforementioned multilateral mechanisms tremendously alleviated the global vaccine inequity.

### 3.3. China’s Vaccine Diplomacy through Marketization in Developing Countries

China has tried to expand the market for its vaccines through diplomacy in developing countries. China based its vaccine diplomacy on its comparative advantages in vaccine R&D, manufacturing and delivery, and it has achieved a relatively significant success [34]. Chinese COVID-19 vaccines have claimed the largest market share in developing countries. A close examination of China’s commitment to COVAX and Gavi helps to shed light on the point that China’s vaccine diplomacy is based on marketization rather than donation. On 12 July 2021, China signed agreements with Gavi, the Vaccine Alliance, to sell 550 million doses of vaccines to Gavi [33]. The overwhelming majority of Chinese vaccines have been provided via sales instead of donations. China has provided most of its vaccines abroad as commercial supplies. China claimed that it would donate additional 10 million doses of vaccines to COVAX [35]. Apparently, the number pales in comparison to that of China’s sales to Gavi in the agreements. Therefore, it is not surprising that China is not among the top donors of vaccines (Figure 1).

Overall, China’s donations of doses have been a small portion of China’s portfolio (see Table 1). China’s vaccine diplomacy through marketization of vaccines can also be observed from the great commercial interests its vaccine companies gained from overseas markets. For example, the sales of Sinovac in 2021 increased to USD 19.4 billion from USD 510.6 million in the prior year. Half of its revenues in 2021 were generated from overseas markets [36].

In addition, China launched vaccine diplomacy to promote extensive partnerships in sales and manufacturing in developing countries to scale up the manufacture of Chinese vaccines overseas. As of 20 April 2022, China has partnered with over 20 developing countries with annual productive capacity amounting to one billion doses, which has significantly enhanced the marketization of China’s vaccines in developing countries [37].

## 4. The Implications of China’s Vaccine Diplomacy for Global Health Governance

China’s vaccine diplomacy has significant implications for global health governance. Given China’s strong capacity in vaccine production and distribution, its vaccine diplomacy makes a huge impact on the landscape of global health governance.

### 4.1. China’s Vaccine Diplomacy Narrowed the Global Immunization Gap

The immunization gap between the developed countries and middle- and low-income countries poses a formidable challenge to global health governance. China regards vaccine diplomacy as an important instrument to narrow the global immunization gap. At the virtual session of the 2022 World Economic Forum (WEF), Xi Jinping stressed the importance of fully leveraging vaccines as a powerful weapon to close the global immunization gap [38]. China’s vaccine diplomacy has significantly contributed to global COVID-19 vaccine equity. According to Wang Yi, China’s Foreign Minister, one in two doses of vaccine administered globally are “made in China”, and China has carried out joint production with 20 countries, with an annual production capacity of one billion doses [39]. China has also repeatedly called on all countries to uphold the primary attribute of vaccines as global public goods to ensure an equitable distribution of vaccines and speed up vaccination to close the gap in immunization [40]. China’s efforts to promote the accessibility and availability helped to narrow the immunization divide. China contributed to global health governance via vaccine diplomacy, as it promoted the accessibility, availability and affordability of vaccines worldwide.

### 4.2. China’s Vaccine Diplomacy Promoted the Right-to-Health Approach to Global Health Governance

The right to health was first articulated as a human right in the Constitution of the WHO. China has attached great importance to the right to health in its domestic vaccine administration. The rights to subsistence and development have been regarded as primary and fundamental human rights in China [41]. During the COVID-19 pandemic crisis, the right to subsistence manifested itself in the right to health.

China adopts a human-right-based approach in its vaccine diplomacy. China’s right-to-health approach in vaccine diplomacy is an extension of its domestic vaccine policy. China regards the right to health as a basic human right. In 2017, China released a white paper titled “Development of China’s Public Health as an Essential Element of Human Rights”. The paper emphasized that “Health is a precondition for the survival of humanity and the development of human society”; “The right to health is a basic human right rich in connotations”; “It is the guarantee for a life with dignity. Everyone is entitled to the highest standard of health, equally available and accessible” [42]. The right-to-health approach has been highlighted in China’s high-level discourses on vaccine policy. The equitable distribution of vaccines is the key to living up to the right to health. Therefore, China has promised to make Chinese-made vaccines a global public good and ensure vaccines are affordable for developing and least-developed countries [43]. “COVID-19 vaccine development and deployment in China, when available, will be made a global public good, this will be China’s contribution to vaccine accessibility and affordability in developing countries,” China’s President Xi Jinping declared at the 73rd World Health Assembly [44]. At the opening ceremony of the Boao Forum for Asia Annual Conference 2021, Xi Jinping reiterated that China would honor its commitment to make vaccines a global public good [45]. China’s right-to-health approach in vaccine diplomacy has not only projected China as a responsible stakeholder in global health governance but also renewed global attention to extraterritorial human rights obligations in global health.

## 5. Challenges for China’s Vaccine Diplomacy Ahead

Undoubtedly, China’s vaccine diplomacy has promoted China’s international image and helped to narrow the vaccination gap between developed and developing countries. However, it faces geopolitical and technological challenges ahead.

### 5.1. Geopolitical Rivalry between China and the U.S.

In an age of escalating great power rivalry, the geopolitical competition between China and the United States has been demonstrated in their vaccine diplomacy. China’s vaccine diplomacy has targeted the developing countries, including in Africa and Latin America. This was regarded as a threat to the dominance of the United States in those areas. The United States accused China of using coercion in making the vaccines available to governments in need [46]. The United States tried to blunt China’s vaccine diplomacy in Latin America by boosting vaccine donations in the region. China’s vaccine diplomacy faced its real test when Biden promised that “America will become the arsenal of vaccines as we were the arsenal of democracy during World War Two” [47]. The European Commission also expressed its concerns over China’s vaccine diplomacy. For example, European Commission President Ursula von der Leyen expressed skepticism over why China has exported its vaccines around the globe while neglecting its own population [48]. China launched vaccine diplomacy to enhance its bilateral relations to boost its influence in eastern Europe, Latin America and the African countries. China’s vaccine diplomacy fits within its agenda of branding itself as a global health leader. In response, the U.S. initiated vaccine diplomacy to shape the international environment to its benefit and tried to claim its global leadership in the fight against the COVID-19 epidemic. The Biden administration views HSR as a clear geopolitical challenge to the United States. To counter China’s ambition to expand both the market share and international influence via vaccine diplomacy, the U.S. partnered with Australia, India and Japan through the Quadrilateral Security Dialogue in March 2021 to finance, manufacture and distribute at least one billion doses of COVID-19 vaccines by the end of 2022 [49]. The Biden administration hosted the Global COVID-19 Summit in 2021 and 2022. China bluntly refused to attend the summit out of concern that China’s attendance might consolidate the U.S global leadership in vaccine diplomacy [50]. Given the geopolitical rivalry between China and the United States, China’s vaccine diplomacy will be challenged by the U.S. with its tremendous vaccine diplomacy.

### 5.2. Concerns over the Efficacy Rate of China’s Vaccines

China’s success in vaccine diplomacy hinges on the efficacy of its vaccines. Biotechnologically speaking, China has made great strides in vaccine research and development to respond to COVID-19, which finds full expression in the fact that the WHO has listed the Sinopharm and the Sinovac-CoronaVac COVID-19 vaccines for emergency use. However, the efficacy of China’s vaccines has been called into question. On 10 April 2021, the director of the Chinese Center for Disease Control and Prevention allegedly said that Chinese vaccines “don’t have very high protection rates” [51]. According to Gavi, the Vaccine Alliance, the efficacy of China’s Sinopharm and Sinovac vaccines in protecting against all symptomatic disease after the second dose is 65–86% and 36–62%, respectively, which is significantly lower than that of Moderna (90–97%) and Pfizer-BioNTech (90–97%) [52]. Some countries also cast their doubts over China’s vaccines. For example, Bahraini officials claimed that it would be offering Pfizer-BioNTech doses to certain high-risk individuals who have already received two Sinopharm jabs [53]. In September 2021, Brazil suspended its use of 12 million shots of China’s Sinovac COVID-19 vaccine [54]. In October 2021, Thailand ceased using the Sinovac COVID-19 vaccine after its supplies were exhausted [55].

The widespread suspicions over the efficacy of China’s vaccine tend to hinder China’s vaccine diplomacy. The Omicron outbreak in China has indirectly demonstrated the low efficacy of China’s vaccines. Since the outbreak of Omicron in China, many cities, including Shanghai, have been locked down. The nationwide lockdown has caused unthinkable damages to China’s economy. However, China seems to have given up on mass vaccination as a viable tool against the pandemic. Mass vaccination has seldom been mentioned by the Chinese government in the fight against the pandemic. It seems to have been excluded from the toolkit of local governments since the outbreak of Omicron in Shanghai. China’s zero-COVID policy through large-scale lockdown instead of mass vaccination indicates that China is not confident in the efficacy of its domestically produced COVID-19 vaccines. Due to widespread concerns over the relatively low efficacy rate of China’s vaccines, China’s exports of vaccines have plunged dramatically so far. For example, China’s top three vaccine producers, Sinopharm, Sinovac Biotech and CanSino Biologics, exported a total of 6.78 million doses in April 2022, a 97% decrease from the peak in September 2021, according to UNICEF [56]. Because of concerns over the relatively low efficacy rate of China’s vaccines, a growing number of countries, including those in southeast Asia, are shifting away from Chinese vaccines [57]. It is hard for China to be the main supplier of the vaccine. The concerns over the efficacy of China’s vaccines makes the future of China’s vaccine diplomacy gloomy.

To improve the efficacy of the COVID-19 vaccines and mitigate the concerns, China has actively invested in vaccine R&D against Omicron. As mRNA vaccines have been more widely used in the world, China has made great efforts to develop its own mRNA vaccines. Sinopharm and Sinovac vaccines targeted at the Omicron variant have been approved to enter the clinical trial [58]. According to Lei Zhenglong, deputy director of China’s National Health Commission’s Bureau of Disease Prevention and Control, China has arranged several R&D tasks in mRNA vaccines. Some with faster timelines are conducting phase III clinical trials abroad, and some are in the process of review and approval. [59]. China’s efforts to develop mRNA vaccines indicate that China expects to mitigate the concerns over the efficacy of its vaccines.

## 6. Conclusions

The COVID-19 pandemic has presented China with a window of opportunity to uplift its international image and influence through vaccine diplomacy. China uses vaccine diplomacy to contribute to its nation branding and present itself in a successful and responsible role against the pandemic [1,60]. China has also been motivated to launch vaccine diplomacy for its comprehensive national security and commitment to building a Health Silk Road for Cooperation. China’s centralized political system makes it easy to adopt a whole-of-government approach to vaccine diplomacy at both the bilateral and multilateral levels. China’s vaccine diplomacy has had a significant impact on the global geopolitical landscape, as China has attempted to turn its health crisis into a geopolitical opportunity. As noted by Yanzhong Huang, “where Beijing’s inoculations go, its influence will follow” [61]. Admittedly, China’s vaccine diplomacy, to some extent, has helped to close the global immunization gap and promote the right-to-health approach to global health governance. However, the sustainability of China’s vaccine diplomacy is questionable. The escalation of the power rivalry between China and the U.S. and the international concerns over the efficacy of China’s vaccines due to biotechnological barriers forebode a gloomy future for China’s vaccine diplomacy.

## Figures and Tables

**Figure 1 healthcare-10-01276-f001:**
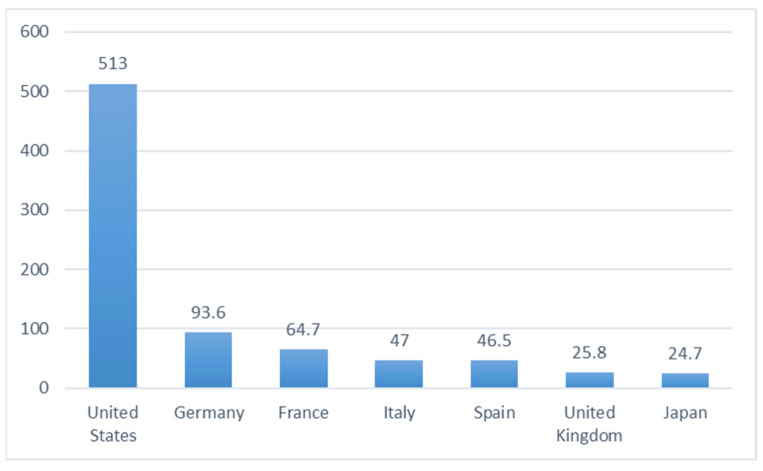
Top 7 countries’ dose donation to COVAX as of 17 January 2022 (unit: million). Source: Gavi, the Vaccine Alliance, “Break COVID Now: The Gavi COVAX AMC Investment Opportunity”.

**Table 1 healthcare-10-01276-t001:** Donated and sold Chinese vaccines by 16 May 2022 (unit: million).

Countries across Regions	Doses Donated	Doses Sold
Africa	74.7	186
Asia Pacific	131	938
Europe	3.6	123
Latin America	12.8	396
Total	222.1	1643

Source: Bridge Consulting, “China COVID-19 Vaccine Tracker”.

## Data Availability

Gavi, the Vaccine Alliance. 2022. Break Covid Now: The Gavi COVAX AMC Investment Opportunity; https://www.gavi.org/sites/default/files/covid/covax/Gavi-COVAX-AMC-2022-IO.pdf. Bridge Consulting. 2022. China COVID-19 Vaccine Tracker; https://bridgebeijing.com/our-publications/our-publications-1/china-covid-19-vaccines-tracker/.

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
