# Peer review of "China’s Vaccine Diplomacy and Its Implications for Global Health Governance"

_healthcare, 2022, doi:10.3390/healthcare10071276_

Round 1

Reviewer 1 Report

Review of “China’s Vaccine Diplomacy and its Implications to Global 2

Health Governance”

The paper argues that China’s pursuit of vaccine diplomacy in regards to Covid-19 vaccines is a result of its holistic policy of biosecurity. It is claimed that China adopts a human rights approach to its vaccine diplomacy.

Comments

1.      Page  1 Line 32 “. China quickly put the pandemic under control domestically…” I think the recent events in Shanghai and Beijing make it clear that no country has put the pandemic under control.

2.      Page 1 Line 27 “China has successfully developed vaccines to tackle the pandemic” I am somewhat concerned with this claim considering the later concerns of efficacy developed later in the paper on the Chinese vaccines delineated on Page 7 lines 299 and 300. I would suggest a less strong sentence.

3.      Page 1 Line 41 “Vaccine diplomacy is not something new. It is as old as the vaccine itself.” Need citation.

4.      In your discussion of why China launched vaccine diplomacy in regards to Covid-19 vaccines you ignore two major strands of the literature. Kobierecka (2022) and Lee (2021) argue that China’s pursuit of vaccine diplomacy is a projection of soft power diplomacy that seeks to overcome the diplomatic setback of the disease arising from within China and expanding on previous soft power projections to enhance the image of China. You should explore how the focus on the holistic policy of biosecurity might encompass the pursuit of overcoming a bad circumstance and projecting a better image or refute their argument. Suzuki and Yang (2022) argue that China’s vaccine diplomacy is primarily driven by its relative advantages in vaccine R&D, manufacturing, and delivery. Again, you should reference this and explore how the holistic policy of biosecurity works with China’s relative advantages in R&D, manufacturing and delivery to produce the kind of vaccine diplomacy it has pursued or refute them.

5.      Table 1 – If China’s approach is holistic, why are its donations so skewed towards Asia? This seems more explained by the projection of soft power argument to me, but I could be wrong.

6.      Line 266-267 “China refused bluntly to attend the Summit for the concern 286

that China’s attendance might consolidate the U.S global leadership in vaccine diplomacy” Need citation.

7.      Need to explain or comment on if China’s policy is holistic what has been the pattern of further R&D on Covid vaccines in China? Has there been attempts at improving efficacy?

Lee, S. T. (2021). Vaccine diplomacy: nation branding and China’s COVID-19 soft power play. Place Branding and Public Diplomacy, 1-15.

Kobierecka, A. (2022). Post-covid China:‘vaccine diplomacy’and the new developments of Chinese foreign policy. Place Branding and Public Diplomacy, 1-14.

Suzuki, M., & Yang, S. (2022). Political economy of vaccine diplomacy: explaining varying strategies of China, India, and Russia’s COVID-19 vaccine diplomacy. Review of International Political Economy, 1-26.

Author Response

Response to Reviewer 1 Comments

Thank you so much for your very insightful and constructive comments and suggestions. Please find the following responses.

  1. Page  1 Line 32 “. China quickly put the pandemic under control domestically…” I think the recent events in Shanghai and Beijing make it clear that no country has put the pandemic under control.

Response: It has been changed to “China has temporarily put the pandemic under control domestically”

  1. Page 1 Line 27 “China has successfully developed vaccines to tackle the pandemic” I am somewhat concerned with this claim considering the later concerns of efficacy developed later in the paper on the Chinese vaccines delineated on Page 7 lines 299 and 300. I would suggest a less strong sentence.

Response: It has been changed to “To some degree, China has successfully developed vaccines to tackle the pandemic.”

  1. Page 1 Line 41 “Vaccine diplomacy is not something new. It is as old as the vaccine itself.” Need citation.

Response: the citation is provided. .

  1. In your discussion of why China launched vaccine diplomacy in regards to Covid-19 vaccines you ignore two major strands of the literature. Kobierecka (2022) and Lee (2021) argue that China’s pursuit of vaccine diplomacy is a projection of soft power diplomacy that seeks to overcome the diplomatic setback of the disease arising from within China and expanding on previous soft power projections to enhance the image of China. You should explore how the focus on the holistic policy of biosecurity might encompass the pursuit of overcoming a bad circumstance and projecting a better image or refute their argument. Suzuki and Yang (2022) argue that China’s vaccine diplomacy is primarily driven by its relative advantages in vaccine R&D, manufacturing, and delivery. Again, you should reference this and explore how the holistic policy of biosecurity works with China’s relative advantages in R&D, manufacturing and delivery to produce the kind of vaccine diplomacy it has pursued or refute them.

Response: The following citations have been added.

Kobierecka and Lee’s articles have been cited to prove that China uses vaccine diplomacy for nation branding at line 342-344 on page 8.

Suzuki and Yang’s article have been cited to stress China’s relative advantages in vaccine R&D, manufacturing, and delivery contribute to its vaccine diplomacy through marketization at line 186-187 on page 4.

  1. Table 1 – If China’s approach is holistic, why are its donations so skewed towards Asia? This seems more explained by the projection of soft power argument to me, but I could be wrong.

Response: In the article, the claim is “China’s pursuit of a Holistic Approach to National Security”. It means that vaccine diplomacy is an important means to achieve its national security. China adopts a whole-of-government approach to vaccine diplomacy and it covers the whole process of vaccine R&D, production and distribution. It does not specifically refer to vaccine distribution geographically.

  1. Line 266-267 “China refused bluntly to attend the Summit for the concern that China’s attendance might consolidate the U.S global leadership in vaccine diplomacy” Need citation.

Response: The following citation has been added

“Foreign Ministry Spokesperson Zhao Lijian’s Regular Press Conference on May 13, 2022, Available online:https://www.fmprc.gov.cn/mfa_eng/xwfw_665399/s2510_665401/2511_665403/202205/t20220513_10685941.html”

  1. Need to explain or comment on if China’s policy is holistic what has been the pattern of further R&D on Covid vaccines in China? Has there been attempts at improving efficacy?

Response: The questions have addressed accordingly in the revised version. China’s attempts to improve efficacy are provided in the revision.

Reviewer 2 Report

It appears to be biased by the researcher.

It seems that the researcher unconsciously interprets the research data to conform to the hypothesis or includes only the data that the researcher thinks are relevant. This can affect the validity and reliability of the research results and can be criticized for lacking scientific precision and transparency.

Of course, it is impossible to eliminate bias completely, but most importantly, there are preventable ways for researchers to reduce bias. All data collected through the study should be analyzed clearly and unbiased, the hypothesis should be reevaluated, and the choice of references should be considered.

Author Response

Response to Reviewer 2 Comments

It appears to be biased by the researcher.

It seems that the researcher unconsciously interprets the research data to conform to the hypothesis or includes only the data that the researcher thinks are relevant. This can affect the validity and reliability of the research results and can be criticized for lacking scientific precision and transparency.

Of course, it is impossible to eliminate bias completely, but most importantly, there are preventable ways for researchers to reduce bias. All data collected through the study should be analyzed clearly and unbiased, the hypothesis should be reevaluated, and the choice of references should be considered

Response: Thanks for your comments and suggestions. More references have been added to reduce bias. The hypothesis has been further evaluated in the revised version

Round 2

Reviewer 1 Report

Acceptable

Reviewer 2 Report

All your notes have been acknowledged and changed.